# A CASE FOR SPARSE POSITIVE ALIGNMENT OF NEURAL SYSTEMS

**Jacob S. Prince** [1] [*], **Colin Conwell** [2], **George A. Alvarez** [1] **& Talia Konkle** [1]
[1] Harvard University, Department of Psychology
[2] Johns Hopkins University, Department of Cognitive Science

## ABSTRACT

Brain responses in visual cortex are typically modeled as a positively and negatively weighted sum of all features within a deep neural network (DNN) layer. However, this linear fit can dramatically alter a given feature space, making it unclear whether brain prediction levels stem more from the DNN itself, or from the flexibility of the encoding model. As such, studies of alignment may benefit from a paradigm shift toward more constrained and theoretically driven mapping methods. As a proof of concept, here we present a case study of face and scene selectivity, showing that typical encoding analyses do not differentiate between aligned and misaligned tuning bases in model-to-brain predictivity. We introduce a new alignment complexity measure – *tuning reorientation* – which favors DNNs that achieve high brain alignment via minimal distortion of the original feature space. We show that this measure helps arbitrate between models that are superficially equal in their predictivity, but which differ in alignment complexity. Our experiments broadly signal the benefit of sparse, positive-weighted encoding procedures, which directly enforce an analogy between the tuning directions of model and brain feature spaces.

## 1 INTRODUCTION

The dominant paradigm in model-brain alignment involves fitting linear encoding models that map from the internal activations of DNNs to responses in visual cortex (Yamins et al., 2014; Kriegeskorte, 2015; Kietzmann et al., 2019; Serre, 2019; Storrs et al., 2021; Conwell et al., 2022). This paradigm has led to impressive progress in our ability to predict response structure along the ventral visual hierarchy. Such advances are especially evident in large-scale, community-wide efforts to "benchmark" the predictivity of DNN models (Schrimpf et al., 2018; 2020; Cichy et al., 2019; Willeke et al., 2022). However, while some progress has been made in attempting to more clearly define the long-term goals of this modeling approach (Doerig et al., 2023; Sucholutsky et al., 2023), it remains unclear whether current methods are the most effective or efficient way to develop DNN models that are increasingly able to explain brain function.

A key issue lies in the basic assumptions of linear encoding, which de-emphasize the particular tuning properties of DNN features, given that they can be remixed arbitrarily to map onto neural measurements. Popular rotation-invariant metrics (such as linear encoding, RSA, CCA, Kriegeskorte et al., 2008; Raghu et al., 2017) pose several challenges. First, in the biological system, the tuning directions of neurons (i.e., their selectivity) is a reliable indicator of the underlying functional mechanism, as shown through extensive neuropsychological evidence linking damage of category-selective cortical areas to category-specific behavioral deficits (Epstein et al., 2001; Barton et al., 2002; Moro et al., 2008, see Kanwisher & Barton, 2011 for review). As such, an over-reliance on rotation-invariant metrics may impair our ability to translate good encoding models into an accurate functional understanding of the neural system (Khosla & Williams, 2023, see also Williams et al., 2021). A related problem is that linear reweighting is itself quite flexible. This complicates the interpretation of whether the predictivity of a DNN is more a consequence of its learned representations, or, of potentially dramatic scaling and rotation of the internal feature space.

---

[*] Corresponding author: jacob.samuel.prince@gmail.com

These issues motivate us to present a series of analyses that argue in favor of stronger, theoretically grounded constraints on linear encoding. First, we propose an intuitive measure for comparing the relative complexity of different alignment methods, which may achieve similar prediction levels through very different feature transformations (**Figure 1A-B**). We perform small-scale simulations showing how this measure can differentiate between different linear models that are equally predictive, by favoring those that place high importance on target-aligned features (**Figure 1C**). Next, we present a case study that shows why rotation sensitivity matters in realistic encoding scenarios. We analyze large-scale human fMRI measurements from the fusiform face area (FFA), comparing two DNN feature spaces that vary dramatically in their theoretical relationship to the target brain region (face- vs. scene-selective units, **Figure 2**). We observe that: (i) typical encoding procedures fail to differentiate between these models; (ii) sparse-positive encoding models do so quite effectively; and, (iii) tuning reorientation provides a quantitative means to understand these different outcomes. Finally, we benchmark several hundred DNNs in their ability to predict face- and scene-selective regions of visual cortex, further validating that the tuning reorientation measure provides a useful way to arbitrate between equally predictive models (**Figure 3**).

## 2 RESULTS

### 2.1 A MEASURE OF ALIGNMENT COMPLEXITY FOR ENCODING MODELS OF NEURAL DATA

We propose a new alignment complexity measure called tuning reorientation ($\Theta$) that can be computed in a straightforward way for a typical neural encoding setup (**Figure 1A**). Tuning reorientation attempts to quantify the rotation of feature vectors towards a target orientation in a multidimensional feature space. It does so by integrating the concept of directional similarity *and* the importance of each feature vector, as determined by the encoding weights themselves (**Figure 1B**).

Computing $\Theta$ for a given encoding target involves several steps. For mathematical description, see Appendix sec. A.1, and for a simple Python code implementation, see Appendix sec. A.2. First, the target vector is normalized to unit length, ensuring that only its direction, not its magnitude, influences the outcome. Each feature vector is also normalized to unit length before computing its cosine distance to the normalized target vector, capturing the angular deviation between each feature and the target. These cosine distances are then weighted by the *absolute magnitude* of each feature weight, integrating the importance of each feature into the measure. This step penalizes high weights placed on misaligned DNN features. Finally, the weighted distances are normalized by the total sum of the absolute feature weights, which adjusts for the overall scale of the weights and ensures comparability across different sets of features. The final measure, $\Theta$, is the sum of these processed distances, providing a scalar value that reflects the weighted angular reorientation of the feature set relative to the target.

**Figure 1C** shows an example of how the tuning reorientation metric compares across a range of encoding scenarios—including Ordinary Least Squares (OLS), Lasso, ElasticNet, and Ridge regression. We evaluate each encoding model under two scenarios: allowing negative coefficients and enforcing positive coefficients only. For illustrative purposes, we consider a 2-dimensional target vector (e.g. a neural response to two different images), and fit each encoding model using the same 100 features. We randomly generate the 100 feature vectors from a standard normal distribution, scaled by $1.5$, and shifted to have a mean of $-0.75$ with a target vector of $[3, 2.5]$.

The visualization makes clear that different encoding models assign weight across the 100 features quite differently, and that the tuning reorientation metric captures these differences. Specifically, when negative coefficients are allowed, many features with poor tuning alignment with the target receive strong weight (yielding relatively large $\Theta$ values, **Figure 1C**, top row). In contrast, the positivity constraint pressures toward solutions emphasizing features that are well-aligned to the target, with positive Lasso achieving a particularly compact solution via minimal reorientation ($\Theta = 0.06$, **Figure 1C**, bottom row). We next present a case for why tuning reorientation is a valuable measure, under more realistic encoding scenarios examining how well DNN features can predict responses in human visual cortex.

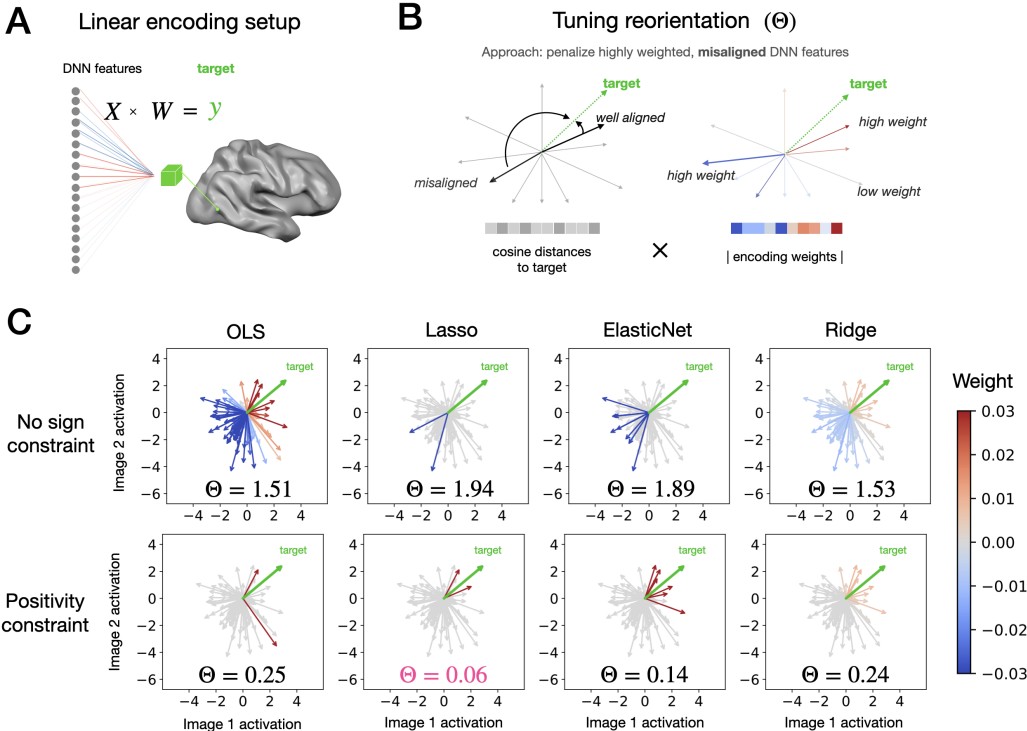

Figure 1: **A measure of alignment complexity for neural encoding.** (A) A typical encoding setup involves mapping from a set of DNN features $X$ to a neural target $y$, via linear weighting ($W$). (B) Tuning reorientation ($\Theta$) quantifies the alignment of a set of feature vectors with respect to a target vector, weighted by the absolute importance of each feature. (C) Comparing tuning reorientation across linear models of a toy dataset (2 images, 100 features), with and without a positivity constraint. Arrows denote different feature tuning directions, and colors represent the weight assigned to each feature in modeling the target vector.

## 2.2 SPARSE POSITIVE CONSTRAINTS PRIORITIZE INTRINSIC TUNING ALIGNMENT

We next perform a targeted case study of the fusiform face area (FFA) using large-scale, high resolution fMRI measurements from the Natural Scenes Dataset (Allen et al., 2022). The FFA is perhaps the canonical example of a brain region with clear empirical evidence linking the feature tuning direction (selectivity to face images over other categories), with functional role (selective involvement in face recognition). An intuitive desideratum for an encoding model is that DNN features used to predict face-selective brain responses should themselves also have face-selective tuning. However, here we show that standard encoding approaches do not respect this correspondence and require additional constraints.

Using an independent DNN "localizer procedure" (Prince et al., 2023), we identify two groups of units with either robust face-selectivity or robust scene-selectivity, in layer fc6 of an AlexNet architecture trained with the self-supervised Barlow Twins objective. Layer fc6 of AlexNet has been analyzed extensively in previous studies of human and primate high-level visual representations (e.g. Bao et al., 2020). For our encoding analysis, we specifically isolate the top 100 most face- and scene-selective units in the layer. **Figure 2A** shows these selective units' tuning orientations when projected into the 2-dimensional PC space of the ImageNet validation set (computed from layer fc6 using a subset of 2000 stimuli). As expected, we observe that face- and scene-selective units tend to orient toward very different parts of high-level latent feature space.

Next, we fit a range of encoding models with different regularization constraints. Critically, we used either the face-selective or scene-selective DNN features to model responses in this face-selective brain region. To assess the goodness of these encoding models, we followed standard procedures and

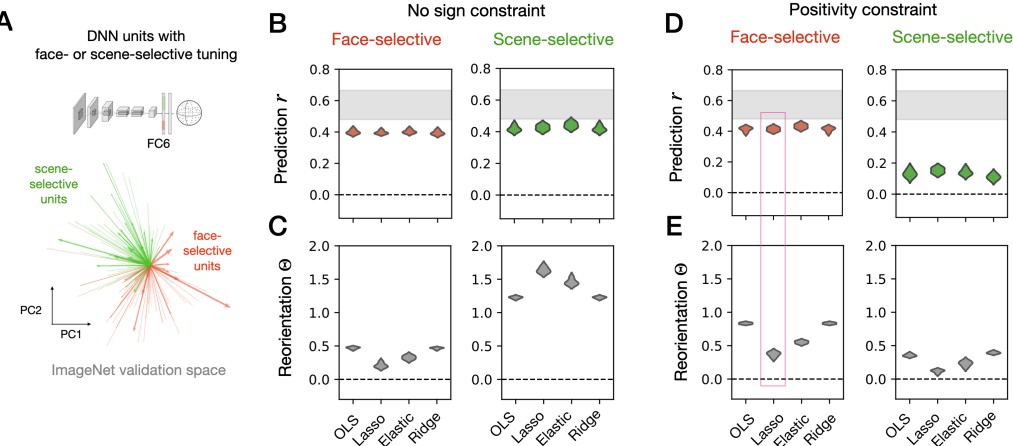

Figure 2: **Encoding analysis of human FFA data.** (A) Face- and scene-selective units are identified in layer fc6 of AlexNet trained on the self-supervised Barlow Twins objective, and their tuning directions are visualized in PC2 space computed over ImageNet validation stimuli. (B-D) Mean encoding scores ($r$) and tuning reorientation ($\Theta$) for 515 test stimuli are plotted for the NSD subjects across different regression schemes, with unconstrained and positive-weighted coefficients. Shaded regions reflect the range (over subjects) of voxel-averaged noise ceilings within FFA.

measured predictivity using a set of held-out images, computed as the average Pearson correlation between the predicted and observed response profiles for each voxel in FFA. We additionally computed our tuning reorientation metric ($\Theta$) for each encoding model, using the DNN feature vectors for the 515 independent test images, and their corresponding model-predicted brain activity matrix as the target. For further details on the encoding procedure, see Appendix sec. A.3.

The results are summarized in **Figure 2B-E**. Surprisingly, we find that both face- and scene-selective units are equally capable of explaining response structure in this face-selective ROI when using standard encoding procedures, across regularization techniques (**Figure 2B**). This is striking, and intuitively problematic, as the two groups of DNN features have dramatically different tuning orientations in image space. However, the $\Theta$ metric provides a means to differentiate these equivalent encoding outcomes – the face-selective units require relatively little tuning reorientation, while the scene-selective units require much more tuning reorientation to predict this face-selective brain region. (**Figure 2C**).

We next considered the same encoding procedures but with an additional positivity constraint on the encoding weights. Now, we find that different brain prediction levels emerge across the DNN feature subsets (**Figure 2D**). The face-selective units yield a similar degree of brain predictivity as before. However, the positivity constraint causes scene-selective units' predictivity to plummet (as high weighting of misaligned features is no longer possible).

Notably, L1-sparse (Lasso) regression appears to stand out among the positive-constrained regression schemes in attaining the best balance of high encoding prediction and low model complexity. In fact, when examining the impact of optimizing the regularization hyperparameter ($\alpha$, see Appendix sec. A.4), we observe that further improvements in performance (and reductions in tuning reorientation) are possible for both Lasso and ElasticNet.

Overall, this case study reveals that standard model-to-brain encoding practices are quite flexible– allowing even oppositely tuned features to be negatively weighted to achieve comparable brain predictivity. We show that measures of tuning reorientation can provide insight into the *alignment complexity* of the encoding model. And, we propose that including sparse positive constraints on the encoding model places greater theoretical weight on direct alignment of tuning directions between the model and brain.

### 2.3 Tuning reorientation differentiates between diverse DNNs in prediction of face- and scene-selective regions

All analyses thus far have focused on a single AlexNet architecture and self-supervised learning scheme. To explore the generality of these effects across larger groups of models, we tested N=106 diverse neural network models, which have been previously analyzed in benchmark analyses of the Natural Scenes Dataset (Conwell et al., 2022). We localized both face- and scene-selective DNN units in these models, and used them to fit standard ridge regression encoding models to brain data from both face- and scene-selective regions. In aggregate, we observed similar brain prediction levels–if anything, face-selective units were generally more predictive of both brain regions. However, our metric of tuning reorientation reveals and quantifies the differences in alignment complexity underlying these comparable encoding fits: mismatched units require more tuning reorientation through the encoding weights than the units with matched tuning. Note that here we focus on FFA and PPA, selecting specific groups of DNN units to demonstrate and validate our approach. However, reorientation is a general metric, and does not require predefined tuning directions.

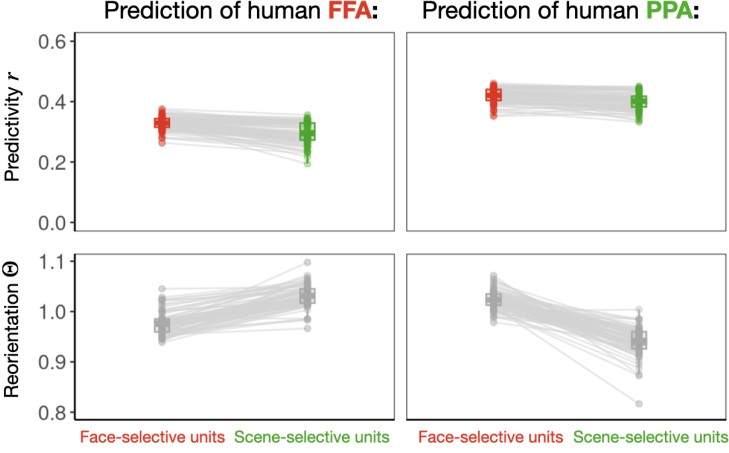

Figure 3: **Assessing face and scene tuning reorientation at scale.** Encoding scores (top) and tuning reorientation values (bottom) are plotted for a group of N=106 DNN models in prediction of human FFA and PPA data. All results involve ridge regression, with no positivity constraint. Dots reflect prediction of face- or scene-selective units from each DNN, with lines connecting results from the same DNN. Encoding fits involve 500 training stimuli and 500 test stimuli.

## 3 Discussion

Here we have shown that standard linear encoding methods enable highly flexible mappings between DNNs and brains, yielding high prediction levels even from DNN units with clearly mismatched tuning to the target brain region. We illustrate this undesirable degree of flexibility through case studies of FFA and PPA, which, despite their functionally distinct roles and selective tuning, can be predicted equally well by face- and scene-selective DNN units. The discovery that such predictions mask underlying differences in tuning orientation leads us to propose a theoretically grounded mapping approach that incorporates both positivity and sparsity constraints.

The positivity constraint effectively limits tuning reorientation, and relates to the operational principles of ReLU networks, where only positive activations propagate to subsequent layers. Allowing negative weights in the encoding model may introduce information into the predictive framework that the subsequent DNN layer itself cannot access. The sparsity constraint further refines the model-to-brain mapping function by reducing feature remixing, thus pressuring toward maximal intrinsic alignment between the tuning of brain and DNN features. Broadly, sparse-positive linking constraints fit within a theoretical framework where individual unit tuning directions are functionally relevant, not arbitrary, for both biological and artificial systems (Prince et al., 2023). Within this framework, models that require less reorientation to map to brain responses are more aligned.

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

## A APPENDIX

### A.1 QUANTIFYING TUNING REORIENTATION

Given a set of feature vectors $\mathbf{F} = \mathbf{f}_1, \mathbf{f}_2, ..., \mathbf{f}_n$, a target vector $\mathbf{t}$, and a set of weights $\mathbf{W} = w_1, w_2, ..., w_n$, the tuning reorientation, denoted as $\Theta$, is computed as follows:

- Normalize the target vector to have a unit norm:

$$\mathbf{t}_{\text{norm}} = \frac{\mathbf{t}}{\|\mathbf{t}\|}$$

- For each feature vector $\mathbf{f}_i$, normalize it to have a unit norm and compute the cosine distance to the normalized target vector:

$$d_i = 1 - \left( \frac{\mathbf{f}_i}{\|\mathbf{f}_i\|} \cdot \mathbf{t}_{\text{norm}} \right)$$

- Weight each cosine distance by the absolute value of the corresponding weight:

$$d_{wi} = d_i \times |w_i|$$

- Normalize the weighted distances by the sum of the absolute weights, to equate the range of reorientation values across encoding scenarios:

$$d'_{wi} = \frac{d_{wi}}{\sum_{i=1}^{n} |w_i|} \quad \text{provided that} \quad \sum_{i=1}^{n} |w_i| \neq 0.$$

- The tuning reorientation $\Theta$ is then the sum of the normalized weighted distances:

$$\Theta = \sum_{i=1}^{n} d'_{wi}$$

### A.2 COMPUTING TUNING REORIENTATION IN PYTHON

```python
import numpy as np

# shape of features should be: (images, DNN units)
# shape of target should be: (images, brain voxels)
# shape of weights should be: (brain voxels, DNN units)

def compute_tuning_reorientation(features, targets, weights):

    # make each feature and target have unit length tuning
    targets_norm = targets / np.linalg.norm(targets, axis = 0)
    features_norm = features / np.linalg.norm(features, axis = 0)

    # compute the cosine distance between each feature and
    # each target tuning. matrix will be (features, targets)
    cosine_distances = 1 - np.dot(features_norm.T, targets_norm)

    # weight each distance value by the absolute weight mapping
    # between the feature and target weights are (targets, features)
    # so must transpose first and then element-wise multiply
    absolute_weights = np.abs(weights)
    weighted_cosine_distances = cosine_distances * absolute_weights.T

    # compute total absolute amount of weight for each target, to
    # standardize the distances.
    # shape of total_weight should be (targets,)
    sum_weight_per_target = np.nansum(absolute_weights, axis=1)
```

```
# Check if the sum of weights is zero to avoid division by zero
if np.nansum(sum_weight_per_target) == 0:
    print("Total weight is zero. Cannot standardize.")
    return np.nan
else:
    # Standardize the weighted cosine distances by the sum of the
        absolute weights, considering each target separately
    weighted_cosine_distances_standardized =
        weighted_cosine_distances / sum_weight_per_target

    # return a vector of reorientation values of shape (targets)
    return np.nansum(weighted_cosine_distances_standardized,
                     axis = 0)
```

### A.3 SUPPLEMENTARY ENCODING METHODS FOR FFA ANALYSIS

The FFA encoding analyses involve the 4 NSD subjects who completed the full experiment, and are repeated in an identical manner for face- and scene-selective units. For each voxel in region FFA-1 (which was defined using the same localizer stimuli as used for the DNN units), we fit 8 linear models (4 regularization methods x positivity-True/False). The regularization methods were OLS, Lasso, Ridge, and ElasticNet. A constant $\alpha$ hyperparameter value of 0.1 was used for all regularization schemes, and the balance between L1 and L2 penalty for ElasticNet was set to the default of 0.5. All features and targets were standardized prior to fitting. Each model fit used data from 1000 subject-specific training images, and the encoding models were then used to predict brain responses to an independent set of 515 subject-overlapping images.

### A.4 OPTIMIZING REGULARIZATION LEVELS FOR PREDICTION OF FFA DATA

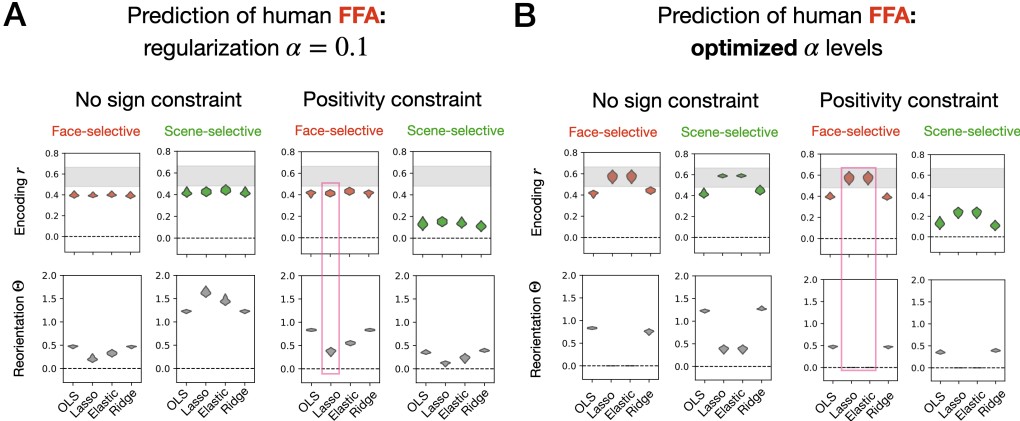

Figure 4: **Impact of optimizing regularization hyperparameters on FFA encoding.** (A) FFA encoding results are plotted for a constant $\alpha$ hyperparameter value of 0.1 (same as Figure 2). (B) Results are plotted after testing a range of 10 logarithmically spaced $\alpha$ values from 0.01 to 10000 for each distinct regression fit, and selecting the value that maximizes brain prediction using an independent validation set of 1000 stimuli. Shaded regions reflect the range (over subjects) of voxel-averaged noise ceilings within FFA.

