# OpenReview forum: "A case for sparse positive alignment of neural systems"
_ICLR.cc/2024/Workshop/Re-Align — ICLR 2024 Workshop Re-Align Poster_

### Official Review · Reviewer_tVsn · 2024-02-22
**What drives model-neural alignment (DNN itself vs. linear mapping to data) and how can this be estimated?**

**Rating:** 3
**Fit:** 3
**Confidence:** 2

**Workshop Review:**

## Summary

This paper investigates the estimation of the alignment of a deep neural network (DNN) model to neural activity (human fMRI, 4 subjects, natural scenes dataset; NSD). The authors suggest to regard any alignment as a two-part procedure composed of (1) representations „from the DNN itself“, and of (2) the „encoding model“ linking the DNN representations to the neural data. They argue that the flexibility of the second step may be so powerful that the inherent DNN tuning properties might be insufficiently reflected in some overall alignment score. To disentangle the relative contributions „from the DNN itself“ and from the „encoding model“, they introduce a tuning reorientation (Θ) measure estimating the alignment of a set of feature vectors (here „from the DNN itself“) to some target vector (here based on fMRI data). Different alignment methods might achieve similar overall alignment scores, but might do so using very different DNN feature transformations, which the tuning reorientation (Θ) might be able to distinguish.

The authors use AlexNet (penultimate layer: fc6) to predict human neural activity from a face-selective region (FFA; fig. 2). They find that both face- and place-selective model units explain activations from cells tuned to faces (FFA) equally well across all regularization methods tested (fig. 2B). To explain this surprising result (intuitively, face-tuned units should be better at predicting activity from face-tuned neurons), they suggest different degrees of DNN feature transformations at play. They measure the relative contribution of the „encoding model“ to the overall fits using tuning reorientation. The authors find that across all type of regularization tested face-selective model units require much less tuning orientation than scene-selective model units (fig. 2C). If, however, they apply a positivity constraint to the weights, the ability of scene-selective model units to predict neural activity from face-tuned neurons is considerably reduced, whereby this is not true for face-selective units (fig. 2D). Further optimizing some regularization hyperparameter can lead to increased alignment levels for Lasso and Elastic (fig. 4B) resulting in minimal tuning reorientation values. Last, only using ridge regression in combination with no positivity constraint in >100 DNNs the authors show how their main results generalize across architectures (fig. 3).


## Strengths and weaknesses
### Strengths
The paper is concise, and clearly written, and has informative, and well-designed figures. It addresses a very interesting problem, namely that model units tuned to category x might predict activity from neurons tuned to category y equally well as model units tuned to category y (fig. 2B). The paper introduces a clearly motivated measure to estimate the contribution of the transformation (through linear mapping) of DNN features to a model-brain alignment score. The introduction of the tuning reorientation (Θ) measure opens a debate as to whether an alignment score should exclusively evaluate the ability of a combination of DNN and linear mapping to predict the data at hand or whether it is also fruitful to consider relative contributions to the overall alignment score of the DNN itself vs. the linear mapping.
Two extremes of this debate might be captured by the following standpoints.

Prediction-focused view: I am only interested in how well a given model (DNN itself + encoding model) predicts some data, and I am agnostic to which model part this ability might mainly be based on (DNN itself or encoding model).

DNN-representation-purist view: I would like to know how well a given model (only DNN itself) was able to predict the data and how much or little fitting (tuning reorientation) was still required from the „encoding model“ (the link between DNN representations and the data) to reach the overall alignment score.
Following the DNN-representation-purist view, a brain alignment score using linear mapping could be divided by the tuning orientation Θ to yield a tuning-reorientation-adjusted version of itself.

### Weaknesses
The tuning reorientation (Θ) measure is well motivated, but I believe it should be normalized by some sort of tuning reorientation (Θ) noise ceiling considering the data at hand. Let’s unpack this by first reiterating why the noise ceiling in the case of prediction r (fig. 2B) is useful. If the prediction r would be expressed relative to the lower bound of the noise ceiling, this would allow straightforward comparisons of prediction r values across regions of interest, e.g. FFA and PPA (see minor requested changes for fig. 3 below). Introducing a tuning reorientation (Θ) noise ceiling in fig. 2B would render the values of tuning reorientation (Θ) interpretable. For example, Θ values within the noise ceiling (indicating a model fit as good as the data to itself) would indicate strong reorientation scores, whereas values above the noise ceiling would indicate poorer reorientation than required to map the data to itself (the theoretical lower bound is 0, so worse Θ values than based on the data itself yield higher reorientation scores).

### Initial recommendation
Accept
- Addresses important question at the heart of brain modeling and opening an interesting debate (for details see „strengths“).
- Suggested measure “tuning orientation” can help to disentangle differential contributions to alignment.

## Requested changes
### Major
Consider creating a tuning reorientation (Θ) noise ceiling as discussed in „weaknesses“.

### Minor
„We perform small-scale simulations showing how this measure can differentiate between different linear models that are equally predictive, by favoring those that place high importance on target-aligned features (Figure 1C).“
- Please add information in this figure showing that the different scenarios are equally predictive. Currently I can only read this in the main text and can see in the figure that the tuning reorientation varies.

### Figure 3
- Noise ceiling missing in upper row. Maybe express predictivity r relative to the lower bound of the respective noise ceiling?
- Perform statistics in upper and lower row to demonstrate whether you do or do not find significant differences between face- and scene selective units in FFA and PPA and for predictivity r and reorientation.

**Reason For Not Giving Higher Score:**

There is no higher score.

**Reason For Not Giving Lower Score:**

Research is original, well executed, and thought provoking.

**Reviewer Domain:**

neuroscience

---

### Official Review · Reviewer_HE3H · 2024-02-22
**important improvement in an alignment metric , weak test of the metric**

**Rating:** 2
**Fit:** 3
**Confidence:** 3

**Workshop Review:**

This work aims to address a significant challenge in the measurement of alignment between two sets of neural features: the commonly used linear mapping is under-constrained. The proposed solution introduces sparse-positive constraints on the mapping. To highlight the differences between constrained and unconstrained mappings, the concept of "tuning reorientation" is introduced. Tuning reorientation measures the rotation of a feature vector towards a target orientation. The alignment between face-selective versus scene-selective units in a Deep Neural Network (DNN) and face-selective areas in humans was measured using both unconstrained and constrained mappings. The results demonstrated that the sparse-positive mapping effectively rejected scene-selective units as a suitable model for face-selective brain areas.

This work tackles an important issue in measuring the alignment of units in DNNs and neurons/voxels in the brain. However, I have a minor concern. While the proposed constrained encoding metric is well-motivated and effective, the test to reveal its efficacy could be more substantiated. The main goal of measuring alignment between models and the brain is to uncover novel insights into the computational processes within the brain, using DNNs as a lens. Applying fMRI localizers to DNNs to define scene versus face-selective units, and validating the proposed metric by assigning higher scores to (face units, face-selective brain region) and lower scores to (scene units, face-selective brain region), seems to miss the broader objective of alignment. This approach presupposes a highly modular processing of different classes of natural stimuli (e.g., faces and scenes) in both the brain and DNNs, an assumption that cannot be taken for granted. Notably, all DNNs evaluated in the study were likely trained on datasets like ImageNet, which are biased towards objects (each image typically features an object with a category label). Thus, scene representations are de-emphasized in favor of objects within the image, a trend observed even in self-supervised models like Barlow Twins. A more appropriate test, assuming modularity, would compare two distinct categories of objects (e.g., faces versus houses).
An even stronger test, before even starting to analyze the alignment to brain regions, would be to measure the alignments between two DNNs with completely known objective functions and identical architecture.
Overall, this work has important implications for the neuroscience community interested in alignments between DNNs and brain regions.

**Reason For Not Giving Higher Score:**

The metric improvement was incremental and, at this stage, does not provide fundamentally novel insights.

**Reason For Not Giving Lower Score:**

Unconstrained linear mapping is a significant challenge in measuring alignments between natural and artificial networks and this work provides a simple yet effective solution to the problem.

**Reviewer Domain:**

neuroscience

---

### Official Review · Reviewer_kQGz · 2024-02-23
**Comparing alignment on models w/ equal brain prediction**

**Rating:** 2
**Fit:** 3
**Confidence:** 2

**Workshop Review:**

The authors compare linear encoding models of fMRI activity to a new alignment approach involving “tuning reorientation” which favors DNNs that achieve brain alignment without distortion of the original DNN feature space. This can help researchers disambiguate between two models which perform similarly in terms of brain prediction but vary greatly in terms of alignment complexity. Using this approach the authors show the benefit of using sparse, positive-weighted encoding models as they lead to easier interpretation between feature tuning and encoding performance.

Tuning reorientation is computed as the sum of normalized outputs after weighting cosine similarities (between brain and DNN features) by the absolute magnitude of the weights matrix produced by the linear regression encoding model.

I do not follow the logic that the “best” encoding model for say, FFA activity, should solely consist of features that maximally activate to face selectivity. The authors support this argument via case study experiments using a subset of scene-selective AlexNet units vs. face-selective AlexNet units to train separate linear encoding models. They find that both encoding models performed similarly well but their tuning reorientation metric showed much better alignment with the face-selective AlexNet model.

I personally do not find it “surprising” that encoding performance was similar for both models. If you have a feature that is 100% anticorrelated with face-selectivity, then it is practically speaking still face-selective: you will know that the lower its activation the more face-specific stuff is going on. Why is it problematic for encoding models to leverage such anticorrelated units? I wouldn’t really call the scene-specific model “more complex” or exhibiting more “distortion” than the face-specific model, the difference between the two is very simple in that one just has opposite tuning. Elaboration on the thinking behind why such tuning reorientation would be useful would be appreciated.

“Allowing negative weights in the encoding model may introduce information into the predictive framework that the subsequent DNN layer itself cannot access.” I did not understand the support for this reasoning.

**Reason For Not Giving Higher Score:**

I am unsure of how impactful it is to be able to identify models with more or less tuning reorientation but equal brain prediction.

**Reason For Not Giving Lower Score:**

Interesting research direction that should prompt good discussion. Concrete results, well-made figures, and valuable discussion points.

**Reviewer Domain:**

cognitive science

---

### Decision · Program_Chairs · 2024-03-02

Accept (Poster)